# Analgesic Effect of Combined Therapy with the Japanese Herbal Medicine “*Yokukansan*” and Electroacupuncture in Rats with Acute Inflammatory Pain

**DOI:** 10.3390/medicines8060031

**Published:** 2021-06-17

**Authors:** Nachi Ebihara, Hideshi Ikemoto, Naoki Adachi, Takayuki Okumo, Taro Kimura, Kanako Yusa, Satoshi Hattori, Atsufumi Manabe, Tadashi Hisamitsu, Masataka Sunagawa

**Affiliations:** 1Department of Physiology, School of Medicine, Showa University, Tokyo 142-8555, Japan; nacci@dent.showa-u.ac.jp (N.E.); nadachi@med.showa-u.ac.jp (N.A.); tokumo@med.showa-u.ac.jp (T.O.); volttarou@med.showa-u.ac.jp (T.K.); kanakotama21@gmail.com (K.Y.); bin7770@me.com (S.H.); tadashi@med.showa-u.ac.jp (T.H.); suna@med.showa-u.ac.jp (M.S.); 2Division of Aesthetic Dentistry, School of Dentistry, Showa University, Tokyo 145-8515, Japan; amanabe@dent.showa-u.ac.jp; 3Department of Orthopedic Surgery, Showa University Fujigaoka Hospital, Kanagawa 227-8501, Japan

**Keywords:** Kampo medicine, electroacupuncture, formalin, phosphorylated ERK

## Abstract

**Background:** Japanese herbal medicine, called Kampo medicine, and acupuncture are mainly used in Japanese traditional medicine. In this experiment, the analgesic effect of *Yokukansan* (YKS) alone and a combination of YKS and electroacupuncture (EA) on inflammatory pain induced by formalin injection were examined. **Methods:** Animals were divided into four groups: a control group, formalin injection group (formalin), YKS-treated formalin group (YKS), and YKS- and EA-treated formalin group (YKS + EA). The duration of pain-related behaviors and extracellular signal-regulated protein kinase (ERK) activation in the spinal cord after formalin injection in the right hind paw were determined. **Results:** The duration of pain-related behaviors was dramatically prolonged in the late phase (10–60 min) in the formalin group. The YKS treatment tended to reduce (*p* = 0.08), whereas YKS + EA significantly suppressed the pain-related behaviors (*p* < 0.01). Immunohistochemical and Western blot analyses revealed that the number of phosphorylated ERK1/2 (pERK1/2)-positive cells and the pERK expression level, which were increased by formalin injection, were significantly inhibited by YKS (*p* < 0.05) and YKS + EA (*p* < 0.01). **Conclusions:** The YKS + EA combination therapy elicited an analgesic effect on formalin-induced acute inflammatory pain.

## 1. Introduction

Acute inflammatory pain caused by trauma or surgery usually disappears as the wound heals; however, prolonged pain with poor prognosis is common. Persistent nociceptive stimuli cause various changes in pain physiology that lead to chronic pain. Chronic pain is characterized by long-lasting enhanced pain perception, including hyperalgesia (a condition in which mild pain is felt strongly) and allodynia (a condition with non-noxious stimuli) [1], and it is one of the health issues to be solved worldwide due to the deterioration of the patients’ quality of life and increased economic loss [2]. Therefore, an early approach to preventing the transition from acute to chronic pain is a reasonable step to reduce the risk of developing chronic pain [1,3].

Japanese herbal medicine, Kampo medicine, and acupuncture are mainly used in Japanese traditional medicine. Physicians in Japan can prescribe both western pharmaceutical and Kampo medicines under the national health insurance system [4]. Previous studies reported the analgesic effects and mechanisms of electroacupuncture (EA), which is a form of acupuncture therapy that combines acupuncture and pulsed electrical stimulation to acupoints [5,6]. However, few studies have investigated the analgesic effects of Kampo medicine [7,8]. *Yokukansan* (YKS) is a Kampo medicine that consists of seven herbs, namely *Atractylodis lanceae* rhizoma (4.0 g), *Poria* (4.0 g), *Angelicae* radix (3.0 g), *Cnidii* rhizoma (3.0 g), *Uncariae cum uncis* ramulus (3.0 g), *Bupleuri* radix (2.0 g), and *Glycyrrhizae* radix (1.5 g) [9]. By using a three-dimensional, high-performance liquid chromatography analysis, 25 ingredients, such as geissoschizine methyl ether (GM) from *Uncariae cum uncis* ramulus and 18β-glycyrrhetinic acid (GA), a major metabolite of glycyrrhizin contained in *Glycyrrhizae* radix, were identified in the methanol fraction of YKS extract [10,11,12]. YKS is used for patients with symptoms such as dizziness, irritability, neurosis, insomnia, and tardive dyskinesia, and infants with night crying and convulsions [13,14,15,16]. There are some reports that the behavioral and psychological symptoms in patients with dementia improved due to YKS use [7,17,18]. Recent clinical investigations and preclinical basic studies indicated that YSK could have an analgesic effect on neuropathic pain [19,20,21]. However, only a few reports have verified the analgesic effect of YKS on acute pain, and the analgesic effect of combined treatments with other traditional medicines have rarely been verified. 

In the present study, we used a formalin-induced pain model, which has been widely used in acute inflammatory pain investigations [22,23,24]. This model is useful for clarifying the mechanisms of underlying persistent pain because formalin injection generates long-lasting mechanical allodynia and hyperalgesia [25,26].

Extracellular signal-regulated kinase 1/2 (ERK1/2) is an intracellular signaling molecule and a member of the mitogen-activated protein kinase family. Activated (phosphorylated) ERK1/2 is associated with important cellular functions, including proliferation, differentiation and migration [27,28], by activating its downstream targets, including CREB and c-fos in neural cells [29]. Dysregulation of ERK1/2 signaling has been shown to develop neuropathic pain [30,31]. ERK1/2 activation in dorsal horn neurons by nociceptive stimuli plays a critical role in central sensitization. The pharmacological inhibition of ERK1/2 activation in superficial spinal cord neurons reduced pain behaviors [32,33], which suggests a correlation between pain behaviors and activated ERK1/2 levels. Therefore, suppressing the activation of ERK1/2 pathway in the dorsal horn neurons is thought to be one of the protective tools for inhibiting the generation and development of neuropathic pain.

In this study, we investigated whether YKS and a combination of YKS and EA have analgesic effects on formalin-induced inflammatory pain in rats and affect ERK1/2 activation.

## 2. Materials and Methods

### 2.1. Animals

Wistar rats (7- to 8-week-old males, purchased from Nippon Bio-Supp Center, Tokyo, Japan) were housed in standard plastic cages in our animal facilities at 25 °C ± 2 °C, with 55% ± 5% humidity, under a light/dark cycle of 12 h/12 h. The rats were provided water and food ad libitum throughout the study duration. This study was approved by the ethics committees of Showa University School of Medicine (chairperson: Masahiko Izumizaki MD, PhD, certificate No. 02082, approved on 1 April 2020). All the procedures of the study were approved by the Committee of Animal Care and Welfare of Showa University and performed in accordance with the committee’s guidelines. The experimental protocol is shown in Figure 1.

### 2.2. YKS Treatment

The dry powdered extracts of YKS (Lot No. 2170054020) used in the present study were supplied by Tsumura & Co. (Tokyo, Japan). The seven herbs comprising YKS were mixed and extracted with purified water at 95.1 °C for 1 h. The soluble extract was separated from insoluble waste and concentrated by removing water under reduced pressure. YKS was mixed with powdered rodent chow at a concentration of 3%. This dose was decided on the basis of the YKS doses found to be effective in our previous study [34,35,36]. The rats not treated with YKS were fed with normal powdered chow only.

### 2.3. Electroacupuncture

The rats were placed in acrylic chambers (width × length × height: 15.0 × 5.0 × 5.5 cm) for half an hour immediately before formalin injection. EA treatment was performed on the 36th (ST36, Zu San Li) acupoint stimulation using an electrostimulator (SEN-8203; Nihon Kohden, Tokyo, Japan). A stainless-steel needle (diameter, 0.20 mm; length, 30 mm; Seirin Co., Shizuoka, Japan) was used for the EA treatment. The ST36 acupoint was located below the knee, on the tibialis anterior muscle, along the stomach meridian (Appendix A) [37]. Acupuncture needles were inserted 10 mm in the muscle layer of the bilateral selected acupoints and connected by their handles to an electrostimulator. EA was performed by passing a square-wave pulse current between the two needles, and the parameters of electrical stimulation were as follows: duration, 0.1 ms; intensity, 15 mA; frequency for 30 min, 4 Hz.

### 2.4. Formalin Test

The animals were assigned to the following groups: (1) control group, (2) formalin injection group (formalin), (3) YKS-treated formalin group (YKS), and (4) YKS- and EA-treated formalin group (YKS + EA). In the formalin-treated groups, formalin (1%, 50 µL; Polysciences, Warrington, PA, USA) was subcutaneously injected into the plantar on the right hind paw while gently restrained in an acrylic box (width × length × height: 15.0 × 5.0 × 5.5 cm). In the control group, the same saline volume was injected in the rats, and YKS administration was started 7 days before formalin injection in the YKS group. In the YKS + EA group, after a week of YKS treatment, EA treatment was applied to the rats 30 min before formalin injection. Formalin injection elicits a marked biphasic pain response, a transient early phase followed by an intense and persistent late phase. Previous studies reported that the response during the early phase was caused by the direct chemical activation of nociceptive afferent fibers whereas, during the late phase, it was involved in the release of inflammatory mediators [38,39]. The animals were transferred to individual test cages immediately after the injection, and the duration of pain-related behavior (shaking, licking, and lifting) was recorded for 60 min, with 12 successive 5-min intervals. Data were evaluated for the cumulative response time, and evaluated separately for the early phase (0–10 min) and late phase (10–60 min) after the intraplantar injection of formalin or saline [40,41,42].

### 2.5. Immunohistochemistry

Sixty minutes after formalin injection, samples of the fifth lumbar (L5) region of the spinal cord were collected for immunostaining. The rats were deeply anesthetized with an intraperitoneal administration of a combination of three anesthetics, namely medetomidine hydrochloride 0.3 mg/kg (Domitor, Nippon Zenyaku Kogyo Co., Ltd., Fukushima, Japan), midazolam 4.0 mg/kg (Sandoz; Sandoz K.K., Tokyo, Japan), and butorphanol 5.0 mg/kg (Vetorphale; Meiji Seika Pharma Co., Ltd., Tokyo, Japan). The rats were intracardially perfused with the anesthetic agents and phosphate-buffered saline (PBS) at pH 7.4, and then with 4% paraformaldehyde in 0.1 M PBS. The spinal cords (L5) were removed and stored overnight in 4% paraformaldehyde solution. The specimens were soaked in 20% sucrose solution for 48 h. The tissues were then subsequently embedded and frozen in a Tissue-Tek optimum cutting temperature (OCT) compound (Tissue-Tek, Sakura Finetek, Tokyo, Japan). The frozen spinal cords in OCT compound were cut into 20-µm sections using a cryostat (CM1860; Leica Biosystems, Nussloch, Germany). The sections were rinsed with PBS three times and incubated with 10% goat serum containing 0.5% Triton X (Sigma-Aldrich Japan Co., Tokyo, Japan) for 2 h for blocking and permeabilization. The sections were then incubated with rabbit anti-phosphorylated extracellular signal-regulated kinase (pERK1/2) antibody (1:500, No. 4370; Cell Signaling Technology, Danvers, MA, USA) overnight at 4 °C and then for 2 h with fluorescence-tagged secondary antibody, anti-rabbit Alexa Fluor 546 (1:1000, No. A31572; Thermo Fisher Scientific, Waltham, MA, USA). After washing with PBS three times, we stained the nuclei with DAPI (4′,6-diamidino-2-phenylindole, 1:2000; Thermo Fisher Scientific) for 10 min. Fluorescent images of the sections were obtained using a confocal laser-scanning fluorescence microscope (FV1000D, Olympus, Tokyo, Japan), and the number of pERK1/2-positive cells in the ipsilateral dorsal horn was manually counted. The mean number of pERK1/2-positive cells was calculated with three sequential sections in each rat.

### 2.6. Western Blot Analysis

After deeply anesthetizing the rats with an intraperitoneal combination of the three anesthetics, they were euthanized, and a spinal cord segment (a right dorsal part of L4–L5) was promptly sampled and treated with liquid nitrogen. The tissues were homogenized with lysis buffer containing 1% sodium dodecyl sulfate (SDS), 20 mM Tris–HCl (pH 7.4), 5 mM ethylene-diamine-tetraacetic acid (pH 8.0), 10 mM sodium fluoride, 2 mM sodium orthovanadate, 0.5 mM phenylarsine oxide, and 1 mM phenylmethylsulfonyl fluoride. The homogenate was then centrifuged at 15,000 rpm for 60 min at room temperature, and the supernatant was collected. We determined the protein concentration using a bicinchoninic acid protein assay kit (Thermo Fisher Scientific) to standardize the sample concentration. The samples (10 μg each) containing the same amount of proteins were subjected to sodium dodecyl sulfate polyacrylamide gel electrophoresis (SDS-PAGE, 10% SDS) and transferred onto a polyvinylidene difluoride membrane. The membranes were blocked with 5% (*w*/*v*) BSA (No. 011-21271, Fujifilm Wako Pure Chemical, Osaka, Japan) for 1 h at room temperature and then incubated with the following primary antibodies overnight at 4 °C: anti-pERK1/2 antibody (1:1000, No. 4370, Cell Signaling Technology) and anti-ERK1/2 antibody (1:1000, No. 9102; Cell Signaling Technology). The membrane was washed with tris-buffered saline buffer with Tween 20 (Sigma-Aldrich Japan Co.) and incubated with the goat anti-rabbit secondary antibody, conjugated with horseradish peroxidase (1:1000, No. 611-1302; Rockland Immunochemicals, Gilbertsville, PA, USA) for 1 h at room temperature. Chemiluminescence images were obtained with a Pierce ECL Western blotting substrate (Thermo Fisher Scientific) and captured with a charged-coupled device camera system (Ez-Capture MG, Atto Co., Tokyo, Japan). The immunoreactivity of each band was quantified using the Lane & Spot Analyzer software (Atto Co.).

### 2.7. Statistical Analyses

All experimental data are presented as mean ± standard deviation (SD). The statistical significance of the differences in the data was evaluated using a one-way analysis of variance in the SPSS version 18 software (SPSS Japan, Tokyo, Japan). The groups were compared using the post hoc Tukey test. *p* values < 0.05 were considered statistically significant.

## 3. Results

### 3.1. Formalin Test

The preemptive analgesic effect of YKS and that of a combination of YKS and EA were examined using a formalin test. After the administration of 1% formalin into the plantar surface of the right hind paw, consistent with previous reports [38,43], two phases of nociceptive responses were identified (Figure 2A,B). In the early phase (0–10 min), the total time spent with pain-related behaviors was significantly increased by formalin injection [F_(3,20)_ = 11.0, *p* < 0.01]; however, no significant differences were found among the three formalin-treated groups (Figure 2A). In the late phase (10–60 min), the duration of the pain-related behaviors was dramatically prolonged in the formalin group as compared with the control group (F_(3,20)_ = 27.9, *p* < 0.01; Figure 2B). The YKS treatment tended to reduce the pain-related behaviors (F_(3,20)_ = 27.9, *p* = 0.080; Figure 2B). The increased pain-related behaviors were significantly inhibited in the YKS + EA group (F_(3,20)_ = 27.9, *p* < 0.01; Figure 2B). During the experimental period, behaviors such as walking and grooming were observed, and body weight was measured daily. No abnormal behaviors were observed in any rats, and there was no difference in the rates of weight gain between the four groups. It is considered that the administration of YKS itself did not affect the onset of pain-related behavior.

### 3.2. Immunostaining of pERK1/2-Positive Cells

To evaluate central sensitization, we examined the phosphorylated (activated) ERK1/2 (pERK1/2) expression 60 min after formalin. The formalin-induced inflammation resulted in the ERK1/2 activation in the superficial dorsal horn on the ipsilateral side of the lumbar enlargement (L5) in Figure 3A,B. The number of pERK1/2-positive cells in the formalin group was significantly increased as compared with the control group (F_(3,16)_ = 41.7, *p* < 0.01). The increase was significantly decreased in the YKS (F_(3,16)_ = 41.7, *p* < 0.05). The same results were seen in the YKS + EA group (F_(3,16)_ = 41.7, *p* < 0.01). The combination treatment with YKS and EA provided a significantly stronger inhibitory effect than the YKS treatment.

### 3.3. Western Blot Analysis of the pERK1/2 Expression

The Western blot analysis also indicated that formalin injection induced an increased expression of pERK1/2 in the spinal cord, and YKS treatment alone inhibited its effect (F_(3,20)_ = 11.6, *p* < 0.05; Figure 4A,B), and that the YKS + EA treatment almost completely blocked the formalin-induced increase in pERK1/2 level (F_(3,20)_ = 11.6, *p* < 0.01; Figure 4A,B).

## 4. Discussion

In the present study, we investigated whether YKS and a combination of YKS and EA have analgesic effects on formalin-induced inflammatory pain and affect the activation of ERK1/2 in the spinal cord. It has been reported that central sensitization in the spinal cord is associated with the development of pain behavior observed in the late phase [38,39,43]. Formalin injection induces increased pERK1/2 expression [44,45], and the inhibitors of ERK1/2 activation suppress the formalin-induced pain behavior [33,44,45]. In this study, the pain behaviors were significantly inhibited by the combination treatment of YKS and EA in the late phase (Figure 2B), and we believe the suppression of ERK1/2 activation is partially implicated in the mechanism of analgesic action (Figure 3 and Figure 4). In addition, because central sensitization is considered to cause chronic pain development [39,43], the combination of YKS and EA may prevent the development of chronic pain. Although the pre-administration of YKS alone was found to have an inhibitory effect on the activation of ERK1/2, it could not significantly suppress the onset of acute inflammatory pain reflected in the late phase. Therefore, clinically, the combination treatment seems more effective for preventing the onset of both acute inflammatory and chronic pain.

Several reports show that YKS is clinically effective against various chronic pain diseases [19,46], and some basic studies have been reported on its mechanisms of action [20,21,47]. The activation of N-methyl-d-aspartic acid (NMDA) receptor, a glutamate receptor, is considered a central sensitization [48]. Meanwhile, Kawakami et al. [49] reported that isoliquiritigenin included in Glycyrrhizae radix, a constituent of YKS, has an antagonistic effect on the NMDA receptor. Furthermore, GM and GA have been reported to ameliorate the dysfunction of glutamate transport into astrocyte in vitro [50,51]. Suzuki et al. [20] showed that the glutamate level in the cerebrospinal fluid in rats with chronic constriction injury is decreased by YKS. Several studies have shown that ERK1/2 activation is elicited by glutamate transmission through NMDA receptors [33,39,52,53]. From these findings, we consider that the blockade of glutamate-mediated neurotransmission with YKS treatment is related to the suppression of ERK1/2 activation.

The mechanism of pain relief by EA has been explained on the basis of various theories [54,55,56,57]. Among these is that low-frequency EA accelerates the secretion of endogenous µ-opioid receptor (MOR) agonists, beta-endorphin, enkephalin, and endomorphin, and attenuates the induction of inflammatory pain [56,57]. Liao et al. [57] showed that, in mice with inflammatory pain, the endogenous MOR agonist released by EA bound to the MOR, preventing the phosphorylation of ERK through the downregulation of protein kinase C activity (an upstream protein of ERK). Furthermore, Kawasaki et al. [58] revealed that treatment with MOR agonists inhibited the ERK activation that was induced by stimulating the C-fiber in the dorsal horn neurons. This suggests that YKS and EA acted synergistically to suppress the activation of ERK1/2.

Some reports have shown that NMDA receptor activation not only increased the neurotransmission after pain stimuli but also decreased the neuronal sensitivity to opioid receptor agonists [59,60,61]. As mentioned earlier, YKS suppresses the activation of NMDA receptors; therefore, pretreatment with YKS may prevent the decrease in MOR sensitivity and enhance the analgesic effect of EA. YKS was administered 7 days before formalin injection in this examination (Figure 1). Several studies revealed that the pre-administration of YKS suppressed opioid analgesic tolerance, but a single dose of YKS had no analgesic effect [62,63]. In our preliminary experiment, a single administration immediately before the formalin injection could not provide any analgesic effect (data not shown). Pre-administration for several days may be required for YKS to suppress the activation of NMDA in vivo. Further research is required to clarify this assumption.

Matsumoto-Miyazaki et al. [64] reported that a combined treatment with the Kampo medicine Kamikihito and acupuncture prevented delirium in patients with cardiovascular diseases who were admitted into intensive care units. Our study and their report indicate that the combined use of traditional medicines, including herbal medicines and acupuncture, might have great latent potential as new treatments for central nervous system disorders.

This study has a limitation. We investigated changes in the duration of pain-related behaviors and ERK1/2 expression levels only in the acute inflammatory pain. As formalin injection induces persistent pain phase [26], we plan to investigate the effect of these treatments in the chronic phase in the future.

## 5. Conclusions

In this study, we showed that the combination of YKS and EA elicited an analgesic effect on acute inflammatory pain induced by formalin injection. Since this combination therapy suppressed the activation of ERK1/2, it may prevent the prolongation of pain and the transition to chronic pain. Furthermore, optimal therapeutic conditions for such combined therapies must be identified by examining the better doses and administration periods of YKS, and more effective frequency, energizing time, and numbers of treatments with EA.

## Figures and Tables

**Figure 1 medicines-08-00031-f001:**
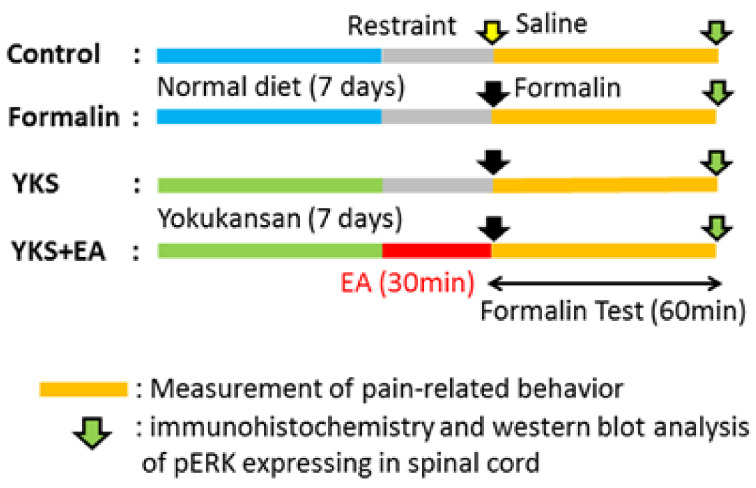
Experimental design of the study. *Yokukansan* (YKS) was mixed with powdered rodent chow at a concentration of 3% and was fed to the rats in the YKS and YKS + EA groups for 7 days. Electroacupuncture (EA) was performed at a frequency of 4 Hz for 30 min immediately before formalin injection in the YKS + EA group. The analgesic effects were determined using a formalin test (for 60 min), and then, spinal cord (L4–L5) samples for immunohistochemistry and western blot analysis were collected.

**Figure 2 medicines-08-00031-f002:**
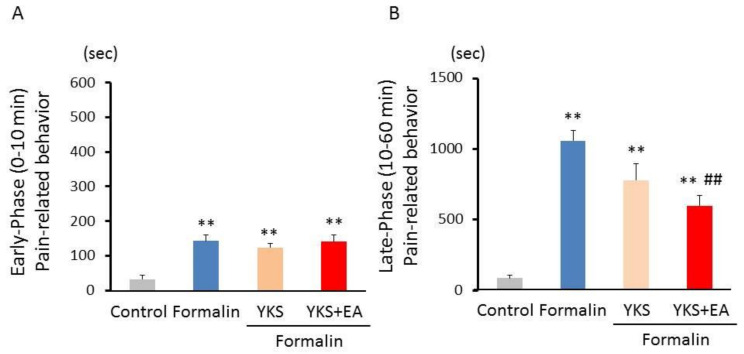
Preemptive analgesic effect of YKS and that of a combination of YKS and EA on pain-related behaviors induced by formalin injection. The total durations of the pain-related behaviors during the early (**A**) and late phases (**B**) are shown. The combination of YKS and EA led to significant antinociceptive effects during the late phase of the response, while YKS alone tended to show analgesic effects. *n* = 6; ** *p* < 0.01 (vs. the control group); ^##^
*p* < 0.01 (vs. the formalin group).

**Figure 3 medicines-08-00031-f003:**
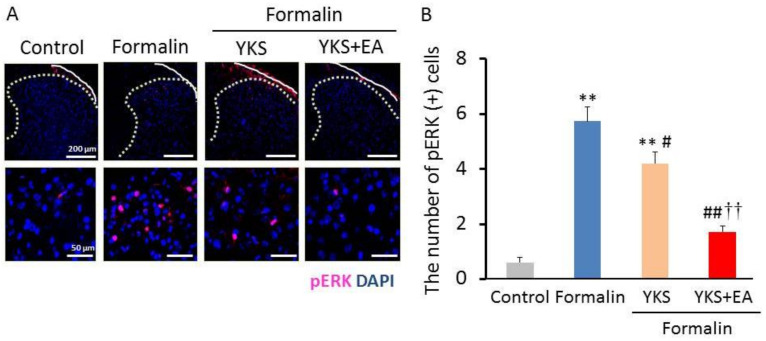
Immunohistochemical analysis of pERK1/2 in the dorsal horn of the spinal cord (L5). Images of pERK1/2 immunoreactivity (red) are shown (**A**, upper; original magnification ×20). White bars = 200 µm. The upper panels are the enlarged images of pERK1/2 immunoreactivity (**A**, lower; original magnification ×60). White bars = 50 µm. (**B**) The number of pERK1/2-positive cells are shown. *n* = 5, ** *p* < 0.01 (vs. the control group); ^#^
*p* < 0.05, ^##^
*p* < 0.01 (vs. the formalin group); ^††^
*p* < 0.01 (vs. the YKS group).

**Figure 4 medicines-08-00031-f004:**
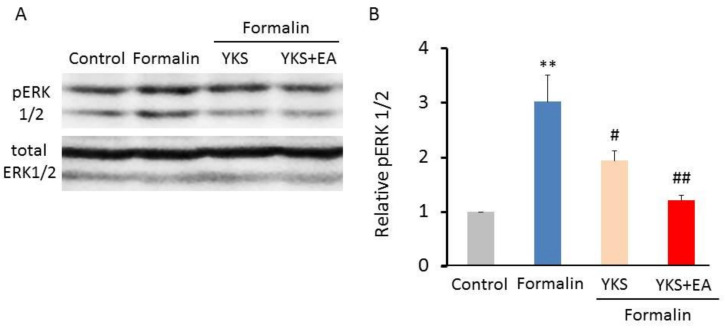
Protein expression levels of pERK1/2 in the spinal cord (L4–L5). (**A**) Immunoblot bands of pERK1/2 and total ERK1/2 are shown. (**B**) Quantified pERK1/2 levels are shown. *n* = 6, ** *p* < 0.01 (vs. the control group); ^#^
*p* < 0.05, ^##^
*p* < 0.01 (vs. the formalin group).

## Data Availability

The data presented in this study are available on request from the corresponding author.

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
