# Peer review of "Analgesic Effect of Combined Therapy with the Japanese Herbal Medicine “*Yokukansan*” and Electroacupuncture in Rats with Acute Inflammatory Pain"

_medicines, 2021, doi:10.3390/medicines8060031_

Round 1

Reviewer 1 Report

  1. Line 23-24, authors stated that “The YKS treatment tended to reduce, whereas YKS + EA significantly suppressed the pain-related behaviors.” There is no statistics provided in the Results (Figure 2?) showing the tendency of pain-related behaviors reduction by YKS treatment. Please clarify or provide statistical analyses or data for this statement. 
  2. Line 24-27, please consider revise the statement to “Immunohistochemical and western blot analyses revealed that the number of phosphorylated ERK1/2 (pERK1/2)-positive cells and the pERK expression level, which were increased by formalin injection, were significantly inhibited by YKS and YKS + EA
  3. Figure 1. illustrates the experimental procedure for YKS+EA not the experimental design of the study. Please provide either an illustration represented for the experimental design or descript the design in text.
  4. Line 137-140, please provide more descriptions for pain realted behaviors accessment and references.
  5. Do you observe any side effect of YKS? For example, Is the food intake of rat affected by YKS? What is the accumulated or daily intake of YKS?
  6. Line 255-258, authors claimed that “Although preadministration of YKS alone was found to have an inhibitory effect on the activation of ERK1/2, it could not significantly suppress the onset of acute pain. Therefore, clinically, the combination treatment seems more effective for preventing the onset of both acute and chronic pain”. It seemed the combination treatment does not effectively prevent the acute pain either according to Figure 2A. Please provide a clarification/justification for this statement.

Reviewer 2 Report

Dear Authors,

The article Ebihara et al. describes “Analgesic Effect of Combined Therapy with the Japanese Herbal Medicine “Yokukansan” and Electroacupuncture in Rats with Acute Inflammatory Pain”. The manuscript presented by the authors is interesting and introduces some new elements. In study, the authors investigated the analgesic effect of Yokukansan (YKS) alone and that of a combination of YKS and electroacupuncture (EA) on  formalin-induced inflammatory pain. Generally, the article written good and solidly. However, it contains a few of minor errors that must be corrected in order for the article to be published in Medicines MDPI.

I suggest not to repeat keywords in the title.

Line 43, 57, 134, 171, 194, 215 - remove the space.

I propose to develop the conclusion.

I recommend publishing in Medicines MDPI.

Sincerely
